# Online assessment of medical students' communication competence in patient encounters: Validation of the VA-MeCo situational judgement test

Sabine Reiser[1,2], Laura Schacht[1], Eva Thomm[1,2], Laura Janssen[3], Sylvia Irene Donata Pittroff[3], Eva Dörfler[4], Kristina Schick[3,5], Pascal O. Berberat[3], Martin Gartmeier[3], Johannes Bauer[1,2]*

**1** Faculty of Education, Educational Research and Methodology, University of Erfurt, Erfurt, Germany, **2** Institute for Planetary Health Behaviour, University of Erfurt, Erfurt, Germany, **3** Department of Clinical Medicine, TUM Medical Education Center, TUM School of Medicine and Health, Technical University of Munich, Munich, Germany, **4** ProLehre | Media and Didactics, Technical University of Munich, Munich, Germany, **5** Medical Faculty and University Hospital Carl Gustav Carus, Institute of Medical Education, TUD Dresden University of Technology, Dresden, Germany

\* johannes.bauer@uni-erfurt.de

## Abstract

### Background

Medical communication is a core task of physicians, and its quality affects both patients and physicians. Hence, the teaching and measurement of medical communication competence (MCC) are crucial during medical studies. This study aims to explore the factorial and construct validity of the *Video-Based Assessment of Medical Communication Competence* (VA-MeCo), an online-situational judgement test (SJT) designed to measure three aspects of medical students' communication competence in patient encounters: advancing content, providing structure, and building relationship.

### Methods

We conducted an online survey with *N* = 395 medical students. *Factorial validity* was tested by confirmatory factor analysis (CFA). To investigate convergent and discriminant aspects of *construct validity*, we tested correlations of participants' VA-MeCo sub-test scores with scores in relevant cognitive variables, patient-interaction, and general personality traits.

### Results

The CFA confirmed the expected three-dimensional factorial structure showing good model fit and highly correlated dimensions. A McDonald's Ω of .94 for the complete test and >.81 for the sub-scales indicated high reliability. Regarding construct validity,

**Data availability statement:** All data files are available from the PsychArchives data repository (DOI: https://doi.org/10.23668/psycharchives.5337). All test materials are available from the OpenTestArchive repository (DOI: https://doi.org/10.23668/psycharchives.14472).

**Funding:** Initials of authors who received the award: SR, LS, LJ, SIDP, ED, KS, POB, MG, JB This work was supported by the German Federal Ministry of Education and Research (Bundesministerium für Bildung und Forschung, BMBF) under Grant [number 16DHB2134]. URL to sponsors' website: https://www.bmbf.de/EN/Home/home_node.html The funders had no role in study design, data collection and analysis, decision to publish, or preparation of the manuscript.

**Competing interests:** The authors have declared that no competing interests exist.

the directions of the correlations were in line with the theoretically assumed associations; correlation sizes partly deviated from our expectations.

## Conclusions

The results support that MCC can be validly measured using the VA-MeCo. The CFA results align with theory and previous studies that proposed three distinguishable dimensions of MCC. The findings on convergent and discriminant validity demonstrate that the test measures MCC as a specific construct that is positively related to patient-interaction; moreover, it can be distinguished from related but more generic constructs (e.g., social competence) and is not overly confounded with construct-irrelevant traits (e.g., intelligence). Notwithstanding the need for further validation, the results indicate that the VA-MeCo is a useful test for assessing MCC in medical education.

## Background

Patient encounters are a central and frequent task of physicians [1,2]. Effective communication between physicians and patients serves as the foundation for successful treatment and the provision of optimal care [3–6]. Research shows that high-quality medical communication is beneficial for both parties [7], fosters patients' well-being, and supports recovery [3,8,9]. Therefore, training medical communication competence (MCC) has become an important—even obligatory—part of medical education in several countries and training as well as assessment methods are receiving increasing research attention [10–13].

Teaching MCC involves the challenge of measuring it as a learning outcome. Reliable and valid assessment tools are required [14–16] not only to gauge learners' progress and provide them with feedback [16] but also to evaluate training interventions [17] and teaching methods [18]. Standardised patient encounters in objective structured clinical examinations (OSCEs) [19] or multiple mini interviews [20] are frequently used and often considered the gold standard in teaching and assessment of MCC [19,20]. However, they are particularly resource-intensive to design and implement. In addition, available MCC assessment instruments differ in focus and in the aspects of communication competence they capture, which limits comparability and standardisation. To address such challenges, there is growing interest in using situational judgement tests (SJTs) as assessment tools for a wide range of competences [21]. In the medical context, SJTs have successfully been developed to measure relevant aspects of communicative and social competences (e.g., empathy [22], communication-related factual and procedural knowledge [23,24]) as well as other characteristics (e.g., safety performance [25]). Previous studies have shown that carefully designed SJTs can yield good reliability and validity [26–28]. Compared with the above mentioned methods, it has been argued that SJTs optimally balance efficient test implementation with validity [28,29].

Despite this growing interest, the number of SJTs developed specifically to assess MCC in medical education remains very limited, and most currently provide only preliminary evidence for their validity. The present study contributes to the further validation of the *Video-Based Assessment of Medical Communication Competence* (VA-MeCo) [30], a computer-administrated SJT designed to measure basic aspects of MCC in medical students. While earlier research provided evidence of the VA-MeCo's curricular and content validity [30], other important aspects, such as its factorial structure and (convergent and discriminant) construct validity have not been evaluated, yet. The present study aimed to contribute to closing this gap.

Below, first, we elaborate on the SJT method and its use for assessment of traits relevant to medical education. Afterwards, we describe the VA-MeCo's test design and the available evidence for its reliability and validity.

## The SJT method and SJTs in medical education

SJTs were originally developed for personnel selection and have continuously gained acceptance as effective measures in other fields [28,29,31]. SJTs present examinees with multiple problem scenarios, which represent typical or critical work-related situations. For each problem scenario, examinees need to evaluate a set of specific response options related to the task at hand [28]. Participants' answers can then be compared with a scoring key, which is often based on expert judgements [32]. Hence, SJTs assess individual abilities to provide knowledge-based and situation-specific judgements in authentic scenarios. That is, they can evaluate an individual's ability to apply professional knowledge to analyse tasks in authentic scenarios, such as clinical situations, and to make judgements about available courses of action [27,33]. Research on competence assessment highlights that such situation-specific skills provide a crucial bridge between individuals' basic dispositions for achievement (e.g., foundational knowledge) and their actual performance in real-world tasks [34]. For modelling the situational context, video-based problem scenarios have proven particularly useful [26,27,35]. In addition to conceptual benefits, such as enhancing authenticity, videos can foster face validity as well as participants' test acceptance and motivation [36,37].

Because of their advantages, SJTs are increasingly used in medical education to measure a multitude of different constructs, such as professionalism [38,39], social skills [40], hygiene competence [41], and preclinical interprofessional collaboration [42]. The use of SJTs in medical education has been associated with favourable psychometric properties [24,43–45]. Moreover, the ease of administration, enhanced authenticity, and high acceptance among participants [22,27,46] make video-based SJTs a promising tool for measuring MCC in an online test setting, especially for large groups [35,43]. SJTs can therefore help bridge the gap between knowledge-based measures of MCC and behavioural assessments, such as OSCEs. By requiring judgements grounded in authentic scenarios, they indirectly capture procedural knowledge and attitudinal dispositions (e.g., empathy, respect, professionalism), thereby contributing to a more comprehensive evaluation of MCC [34].

## The VA-MeCo test of medical communication competence

**Conceptual basis and test structure.** The VA-MeCo is a video-based online-SJT for assessing MCC in medical education with a theoretical basis in (medical) communication theory and established curricular standards of patient-physician communication [47–50]. The Calgary-Cambridge Guide [48] guided the selection and design of the tasks and the setting of quality standards for the test items. Moreover, the test structure draws upon the *Munich Model of Professional Conversation Competence*, an empirically validated model that has been used before to guide curriculum and assessment development in medical education [49,50]. The Munich Model of Professional Conversation Competence conceptualises professional communication competence as a hierarchical, multidimensional construct where the top-level factor of general medical communication competence subsumes three correlated sub-dimensions: (i) facilitating joint-problem solving, (ii) organising conversations in a proactive and transparent way, and (iii) fostering a strong working relationship. This dimensional structure has been empirically corroborated in several studies [50,51].

The VA-MeCo was designed to represent this theoretical structure, to allow a differentiated assessment of MCC. The test measures MCC by three scales: (i) advancing the content of the conversation, (ii) structuring the conversation, and (iii) building a relationship with the patient. The advancing content-scale is based on the joint-problem solving-dimension of the Munich Model of Professional Conversation Competence [49,50]. Both emphasise the importance of a shared understanding between patient and physician. However, the focus of the advancing content-scale is less on problem solving and more on the progression of the conversation, as described below.

The *advancing content*-scale assesses participants' ability to advance the content level of the conversation effectively, for example by gathering information, explaining and planning [48,52,53]. The *providing structure*-scale addresses examinees' ability to structure a conversation actively by guiding the patient through the conversation in a comprehensible and systematic manner. This involves the systematic use of meta-communication, such as providing summaries and managing transitions [48,52,53]. Finally, the *building relationship*-scale refers to the relational dimension by addressing participants' ability to establish and maintain good working relationships with patients throughout the conversation. This includes aspects such as empathy and consideration of the patient's needs [48,52,53]. Hence, on the basic level, the test delivers sub-scale scores on the three basic dimensions of MCC. These sub-scales can be aggregated to a total score representing the top-level construct of MCC. Therefore, depending on the intended purpose of the assessment, the VA-MeCo allows a global assessment of participants' total achievement as well as detailed reports concerning their differential achievement on the three sub-dimensions (e.g., for providing feedback).

**Test tasks and procedure.** The VA-MeCo's test tasks have a standardised structure and focus on critical parts of physician–patient communication. S1 Fig in the online supplementary material shows an example. Each task begins with a written description of the scenario that provides a background of the patient and the general situation. All scenarios focus on the medical history taking during an initial encounter between a physician and a patient. This initial description is followed by a short video sequence of the encounter (~ 60 s) that stops at a critical point in the conversation. The respondents are then provided with a medical communication goal that the physician wants to achieve as well as 3–5 answer options featuring statements to continue the conversation.

The participants' task is to assess the effectiveness of *each* answer option in achieving the stated communication goal on a 6-point rating scale (1 = *very ineffective*, 6 = *very effective*), with the answer options differing in their effectiveness. These ratings cover all three basic dimensions of MCC, so that participants judge each answer option in terms of its effectiveness in advancing the content, providing structure, and building or maintaining a positive relationship. Hence, the test has a nested structure comprising 11 patient encounter scenarios ("*tasks*"), each featuring 3–5 embedded *answer options,* and three *ratings* per option. These ratings constitute the test items (overall 117 items).

Test scores are obtained by comparing participants' effectiveness ratings with an expert- based key using so-called *raw consensus scoring*, a common method of scoring SJTs [54]. The scoring key was developed in a previous study with a sample of experts in medical communication training who demonstrated strong rater agreement [30]. Performance scores are calculated for each item by determining the squared difference between the participant's answer and the respective expert rating [55]. The scores are then reversed so that higher values express higher levels of competence, and they are summed up per sub-dimension.

**Existing evidence on psychometric quality criteria.** Previous studies provided preliminary evidence on the VA-MeCo's usability, reliability and content validity based on two expert studies, cognitive pretesting, and a preliminary test evaluation based on a study with medical students (*N* = 117) [30]. Results suggested good reliability for the total test score (Cronbach's α > .93) and sub-test scores regarding the three dimensions of MCC (α > .83). The two expert studies supported curricular and content validity by confirming the medical correctness of the content and relevance for measuring medical students' MCC. Finally, both expert and student data supported the high acceptance and perceived usability of the VA-MeCo as an assessment tool. Although these data provided initial supportive evidence, further psychometric evaluation is required.

## Validation strategy of the present study

**General approach.** This study aimed to gain additional evidence on the VA-MeCo's psychometric properties with a focus on (a) *factorial validity* and *reliability* and (b) *convergent and discriminant construct validity*. According to the standards for educational and psychological testing [56], collecting evidence on different validity aspects is essential to argue for the interpretability of test scores. Evaluating the mentioned validity aspects enhances the available evidence that the VA-MeCo measures empirically discernible and theoretically relevant dimensions of MCC.

a) *Factorial validity*: We sought to test whether the empirical structure of the test scores reflects the theoretically assumed three-dimensional structure of MCC. Given these a-priori assumptions, confirmatory factor analysis (CFA) was the appropriate method evaluating factorial validity [57,58]. Because issues related to dimensionality are closely related to reliability [59], we also examined whether the good test reliability found in a previous study [30] was replicated in the present study.

b) *Construct validity*: To assess convergent and discriminant validity, we followed the classical construct validation approach, which involves examining a network of correlations between test scores and other variables [57,60]. According to this approach, construct validity is supported when the empirical correlations of the test scores match theoretical expectations. For convergent validity, we tested expected associations of the three VA-MeCo sub-scales (i.e., advancing content, providing structure, and building relationship) with similar measures and criterion-related external variables; for discriminant validity, we tested whether correlations with expectedly unrelated or distal measures were indeed low (see below). In these analyses, we focussed on the VA-MeCo's sub-test scores rather than the overall MCC-score to gain a more differentiated evaluation of convergent and discriminant validity.

**Investigated measures and hypotheses for construct validation.** For investigating construct validity, we selected relevant validation variables by considering, on the one hand, substantive research on MCC [3,6,49,61], and general research on the SJT method (e.g., [29,62]) on the other hand. Table 1 presents the selected variables categorised in (i) domain-specific and generic *cognitive variables* (i.e., prior knowledge and experience; intelligence), (ii) variables relevant to *patient-interaction* (i.e., empathy; patient orientation), and (iii) *general personality traits* (i.e., social competence and Big Five personality traits). To measure these constructs, we employed well-established existing instruments that have a sound psychometric basis (see *Measures*). The assumed correlations between the VA-MeCo's sub-scales are also summarised in Table 1 and elaborated below.

a) *Cognitive variables:* Working on the VA-MeCo's tasks requires knowledge-based judgement of the answer options in relation to the specified communication goal. Therefore, we assumed positive convergent correlations between the participants' test scores and their *prior knowledge and experience* in medical communication as well as with *study progress (i.e., semester)*. Moreover, we assumed that there are, at maximum, medium-sized correlations with *intelligence*. While intelligence is a distal construct to MCC, meta-analysis show that performance on SJTs in general has a low-to-moderate correlation with intelligence, with an estimated mean population correlation of $\rho = .32$ [26]. Hence, correlations of the VA-MeCo's sub-tests with intelligence $r \leq .3$ provide evidence for discriminant validity.

b) *Patient-interaction variables*: A good working relationship with the patient as a central dimension of MCC is a crucial aspect of patient-oriented communication. The concept of patient centredness/patient orientation implies an egalitarian doctor–patient relationship, which means shared power and responsibility between both parties involved [70]. Another important aspect of patient-centred communication is an empathic response to the emotional needs of patients [71,72]. Hence, we assumed positive convergent correlations with *empathy* and the building relationship dimension, but no correlations with the advancing content dimension and providing structure dimension. For the association between test performance and *patient orientation*, we expected positive correlations.

**Table 1. Summary of external variables, instruments, and hypotheses for evaluating construct validity in relation to VA-MeCo.**

| Variable | Sub-scales | Hypotheses on correlations with the VA-MeCo |
|---|---|---|
| *Cognitive Variables* | | |
| Prior knowledge and experience [63] | MCC-courses taken, study progress | Positive correlations (*r>0*) |
| Intelligence (DESIGMA A+ [64]) | — | At maximum medium correlations (*r≤.3*) |
| *Patient-interaction variables* | | |
| Empathy (IRI [65]) | Perspective taking, emotional concern | Positive correlations with *building relationship* (*r>0*); zero or negative correlations with *advancing content* and *providing structure* (*r≤0*) |
| Patient orientation (PPOS-D12 [66]) | Sharing, caring | Positive correlations (*r>0*) |
| *General personality traits* | | |
| Social competence (ISK-K [67]) | Social orientation, offensiveness, self-organisation, reflexibility | At maximum low positive correlations (*r<.1*) |
| Personality (BFI-K [68]) | Extraversion, agreeableness, conscientious-ness, neuroticism, openness | At maximum low positive correlations (*r<.1*) |

DESIGMA A+ = Design a Matrix – Advanced +; IRI = Interpersonal Reactivity Index; PPOS-D12 = Patient Provider Orientation Scale – German version; ISK-K = Inventory of Social Competences – short version; BFI-K = Big Five Inventory short version. Statements about the expected sizes of the correlations refer to Cohen's [69] standards ($r \sim .1$ = low; $\sim .3$ = medium; $\sim .5$ = large). Note that we made predictions about the sizes of the correlations only when doing so could be based on sufficient evidence from prior research.

c)  *General personality traits*: Both communication skills in general and MCC in particular, are content- and domain-specific [73]. This implies that MCC is a more specific construct than general social competence, even though the two are certainly related. Social competence is defined as a person's knowledge, skills and abilities to promote the quality of their own social behaviour [74]. Hence, social competence helps individuals achieve their goals in specific situations while maintaining social acceptance [74]. Given their different scopes, we assumed that low positive correlations of the VA-MeCo's sub-scales with *social competence* would provide discriminant evidence that MCC is discernible from general social competence.

Communicative behaviour is also influenced by general personality traits, such as agreeableness, openness, and con-scientiousness [75–77]. Moreover, there is evidence that the Big Five personality traits have a general impact on partici-pants' performance in SJTs, regardless of the specific trait being assessed. The meta-analysis by McDaniel et al. reported average correlations with conscientiousness ($\rho = .27$), emotional stability ($\rho = .22$) and agreeableness ($\rho = .25$), and smaller correlations with extraversion ($\rho = .14$) and openness to experience ($\rho = .14$) [26]. Hence, we assumed that, at maximum, low positive correlations of the VA-MeCo's sub-tests with the Big Five personality traits would provide evidence for dis-criminant validity.

## Methods

### Participants

This study was implemented as an online survey with medical students, including all phases of medical education and was conducted between June 25, 2020, and December 1, 2020. The study was approved by the medical ethics com-mission of the Technical University of Munich [14/20S]. The students' participation in the study was voluntary. They were informed in advance and explicitly provided informed written consent. Participation took approximately 1.5 hours on average, with an average of 45 minutes for completing the VA-MeCo. Only participants who provided informed consent, satisfied a minimum time on task-criterium, and completed at least five tasks of the VA-MeCo were included in the final

analysis. No other data exclusions were made. Note that due to the decentralised recruitment strategy, the amount and reasons for non-participation could not be determined.

The final sample consisted of $N = 395$ medical students, of whom 70.33% were female, 29.04% male, and 0.63% diverse. This gender distribution aligns with official statistics for German medical students [78]. The sample covered all phases of German medical education, albeit with a focus on the pre-clinical and clinical phases where medical communication courses predominantly take place (age: $M = 23.47$, $SD = 3.86$; semester of study: $M = 5.77$, $SD = 3.55$; stage of study: 41.12% pre-clinic, 51.60% clinic, 6.57% practical year, 0.71% other). The resulting sample size enabled testing correlations as low as $r = .15$ with more than 90% statistical power in the analyses on construct validity and is also sufficiently large for the CFAs [58].

### Measures

All participants first completed the VA-MeCo, followed by the questionnaires and achievement tests related to the validation variables. The study was conducted individually online: participants received written instructions at the beginning and then were guided through the procedure by the survey software, without personal assistance. Instruments for the validation variables were chosen to align conceptually with our research objectives, focusing on well-established measures that have been psychometrically evaluated and supported by prior research. A short description of the selected instruments, along with their reliabilities, is provided in Table S1. For more detailed information, readers are referred to the original publications and test documentations (see references in S1).

### Analysis

General statistical analyses were done using IBM SPSS 28 and CFA-modelling in the Mplus 8.3 software [79] using robust Full Information Maximum Likelihood estimation. To reduce the CFA's model complexity due to the abovementioned nested test structure (i.e., ratings of answer options nested in test tasks), we aggregated items per task for each of the three dimensions. Thus, the CFAs represent the test structure at the task level. We proceeded by first estimating separate unidimensional CFA models per respective MCC dimension (M1–M3) and then combining them in a three-dimensional model (M4). Because each task was rated on all three dimensions of MCC this final model contained expected residual correlations of the same tasks across the three dimensions (for a path diagram, see S2 Fig). Reliability was assessed using McDonald's ω, which is preferable to Cronbach's α [59] and can be interpreted in the same manner. Construct validity was analysed using Pearson correlations.

## Results

### Factorial validity

The three separate unidimensional CFAs for the VA-MeCo sub-scales (M1–M3) resulted in good model fit, as evidenced by the $\chi^2$-tests and standard fit indices (Table 2). For the dimensions advancing content (M1) and building relationship (M3), the $\chi^2$-tests were non-significant despite the large sample size. All other fit indices were good. For the providing structure dimension (M2), the $\chi^2$-test was statistically significant, pointing at some degree of misfit; however, all other fit indices hinted at good model fit.

Based on these results, we proceeded with the estimation of the combined three-dimensional model of MCC (M4). Again, this model showed an overall acceptable model fit despite the somewhat diminished value of the comparative fit index. The standardised factor loadings were of a substantial size (i.e., ~ .4 or larger; [59]), with only Task 5 having a relatively low loading on the content dimension (see S2 Table).

The results from M4 revealed substantial positive correlations among the three dimensions ($r \geq .8$; S2 Table). Given these high correlations, one might argue that a one-dimensional model (i.e., a single general factor of MCC) would provide

**Table 2. Model fit of the CFA models.**

| Model | χ²-test | RMSEA [90% CI] | CFI | SRMR |
|---|---|---|---|---|
| *Unidimensional models for the VA-MeCo sub-scales* | | | | |
| M1 Content | $\chi^2(44) = 56.802$, $p = .093$ | .027 [.000,.046] | .971 | .045 |
| M2 Structure | $\chi^2(44) = 64.690$, $p = .023$ | .035 [.013,.052] | .967 | .050 |
| M3 Relationship | $\chi^2(44) = 59.609$, $p = .058$ | .030 [.000,.048] | .977 | .050 |
| *Complete models of MCC including all VA-MeCo sub-scales* | | | | |
| M4 Three-dimensional model | $\chi^2(459) = 733.697$, $p < .001$ | .039 [.034,.044] | .921 | .065 |
| M5 One-dimensional model | $\chi^2(462) = 867.470$, $p < .001$ | .047 [.042,.052] | .883 | .066 |

RMSEA = root mean square error of approximation; CFI = comparative fit index; SRMR = standardised root mean square residual; CI = confidence interval; MCC = medical communication competence.

a more parsimonious representation of the data. To test this, we compared M4 to a one-dimensional model (M5), in which all tasks loaded onto a single MCC factor. For completeness, we also evaluated three two-dimensional models (M6–M8), each combining two sub-scales into a single factor (e.g., one factor for relationship and another one combining content and structure). The results of the χ²-difference tests indicated that M4 provided a significantly better fit than all alternative models (S3 Table). This was corroborated by information criteria that consistently showed M4 as the best-fitting model among the candidates. Therefore, we selected M4 as the final model.

In the final step, we analysed the reliabilities of the test's sub-scores and total score, which proved to be high throughout (total score: ω = .94; advancing content: ω = .82; providing structure: ω = .88; building relationship: ω = .88).

## Construct validity

The Pearson correlations of the investigated measures with the three dimensions of MCC are listed in a heatmap in Fig 1. Regarding the cognitive variables, there were significant positive correlations of all VA-MeCo sub-scales with *prior knowledge and experience* in the field of medical communication, as well as with *study progress,* which is in line with our hypothesis. Moreover, we found low positive correlations with *intelligence*, which supports our expectations.

Regarding the patient-interaction variables, as expected, both sub-scales contained in the *empathy* measure had higher correlations for the VA-MeCo's relationship sub-scale than with its content and structure sub-scales. In addition, corroborating our hypothesis, there were significant positive correlations of all VA-MeCo sub-scales with the sharing and caring scales contained in the *patient orientation* instrument. However, these correlations were somewhat below the expected value of .30 for the content dimension.

Concerning the general personality traits, the correlations for the three sub-scales of the ISK-K as a measure of *social competence* ranged from low negative ones for offensiveness and reflexibility to low positive correlations for social orientation and self-organisation. These findings are partially in line with our hypothesis. For social orientation, the correlations were, as expected, in the range of positive to small-sized associations, acknowledging that the correlation with building relationship was marginally higher than that with the other VA-MeCo sub-scales. For reflexibility, the correlations were even lower and not significantly different from zero. The correlations with self-organisation were somewhat higher than expected, particularly for building relationship. Finally, unlike expected the correlations with offensiveness were small and negative, although not significantly different from zero.

For the five sub-scales of *the Big Five personality traits*, we found correlations close to zero for extraversion and neuroticism. For agreeableness, conscientiousness, and openness, there were low positive correlations for each of the three

| Variable | Sub-scales | Content | Structure | Relationship |
|---|---|---|---|---|
| Prior knowledge | | .16** | .15** | .19** |
| Semester | | .25** | .29** | .33** |
| Intelligence | | .09 | .16* | .25** |
| Empathy | Perspective Taking | .03 | .03 | .15** |
| | Emotional Concern | .08 | .07 | .17** |
| Patient orientation | Sharing | .25** | .31** | .42** |
| | Caring | .23** | .32** | .50** |
| Social competence | Social Orientation | .08 | .09 | .16* |
| | Offensiveness | -.13 | -.13 | -.13 |
| | Self-Organisation | .16* | .17* | .26** |
| | Reflexibility | .00 | -.04 | .05 |
| Personality | Extraversion | -.05 | -.04 | .01 |
| | Agreeableness | .09 | .10 | .16** |
| | Conscientiousness | .12* | .09 | .13* |
| | Neuroticism | .00 | .01 | .01 |
| | Openness | .04 | .06 | .14* |

**Fig 1. Heatmap for Pearson correlations of the measures with the three dimensions of MCC.** The correlations in boldface are in line with the stated hypotheses (see Table 1). *=p ≤ .05 (two-tailed). **=p ≤ .01 (two-tailed).

dimensions, especially for the building relationship dimension. As with social competence, these findings are consistent with our hypotheses in terms of the sign of the associations, although partly not in terms of size.

## Discussion

This study aimed to extend the existing validity evidence for the VA-MeCo, a video-based SJT of MCC [30]. To this end, we first focused on *factorial validity* and investigated whether there is evidence that the empirical structure of the test scores reflects the theoretically assumed one and whether the resulting scales are sufficiently reliable. Second, we investigated *construct validity* by testing a net of assumed associations among the VA-MeCo sub-scales and a set of convergent and discriminant measures following the classical construct validation approach [56,57].

Regarding *factorial validity*, the CFA results corroborated that the test adequately reflects the underlying theoretical model of MCC and aligns with previous findings on its dimensional structure [49,50]. Our results also replicated preliminary evidence of the VA-MeCo's strong reliability [30]. Although the high inter-factor correlations might be seen as a limitation, they are consistent with the hierarchical structure of MCC as described in the *Munich Model of Professional Conversation Competence*, where the three dimensions form a higher order construct of general MCC [50]. We deliberately chose not to test a hierarchical CFA model with a second-order MCC factor because, in general, a second-order factor model with three first-order factors is statistically equivalent to a model with three correlated factors (cf. M4) [58]. Since such models cannot be distinguished statistically, estimating an additional hierarchical model would not have provided meaningful insights. Moreover, the comparisons with the alternative models (M5–M8) provide further evidence that the three dimensions are not only theoretically but also empirically discernible aspects of MCC. Thus, test result interpretation can focus both on the overall MCC score as a comprehensive measure and the respective sub-scale scores. The latter may be preferable, for instance, for providing detailed feedback to learners [80]. Although the high inter-factor correlations suggest that the dimensions are closely related, they do not imply redundancy. Rather, each sub-scale captures a distinct yet interconnected facet of MCC, allowing for nuanced feedback that can highlight specific strengths and areas for improvement [38].

Regarding *construct validity*, we tested hypothesised correlations with measures of (a) cognitive variables as well as (b) patient-interaction and (c) more general personality traits. For *cognitive variables*, significant positive correlations with prior knowledge and study progress provided convergent evidence that the VA-MeCo differentiates between groups with different levels of MCC. Overall, correlations with prior knowledge were notably smaller than those with study progress. Although we based our questions on these variables on prior research, we acknowledge that they are most likely imperfect proxies for students' actual prior knowledge. Regarding discriminant validity, the identified low positive correlations with intelligence suggest that test performance does not depend excessively on intelligence. As discussed above, meta-analysis indicates that SJTs exhibit, on average, medium-sized correlations with intelligence [26]. The observed correlations between the VA-MeCo's sub-scales and intelligence were lower than expected based on this research. Therefore, we conclude that, although test performance in the VA-MeCo depends on intelligence to some degree—as is likely the case with any standardised achievement test—the measurement of MCC is clearly distinguishable from intelligence.

In terms of *patient-interaction variables*, significant correlations with empathy and patient orientation—especially regarding the building relationship dimension—were found, as expected. This outcome provides evidence of convergent validity. These correlations align with earlier research emphasising that a crucial aspect of patient-centred communication is displaying empathy in response to the emotional needs of patients [70,81]. Furthermore, these positive correlations reflect that building a good working relationship with the patient is a vital aspect of patient-centredness [70,82]. They are also consistent with findings on the importance of an egalitarian doctor–patient relationship, which focuses on the patient, for facilitating effective conversations between patients and physicians [70].

With respect to *general personality traits*, some of the observed correlations with social competence differed to some degree from our expectations. We had assumed at maximum small positive correlations (i.e., $r \leq .1$). Results showed that the correlation between social orientation and the building relationship sub-scale was slightly higher than expected but could still be considered essentially small. This finding might be explained by the fact that people with a high social orientation generally possess a more positive attitude towards others and, thus, are better at empathising and socially connecting with others [67]. Additionally, unlike expected, small negative correlations were observed for offensiveness and all three VA-MeCo sub-scales. However, these correlations did not differ significantly from zero and thus pose no threat to discriminant validity. One potential reason for the descriptively negative correlations may be that people with high offensiveness effectively advocate for their own interests, which is not in line with a patient-centred orientation [67]. Finally, the correlations for self-organisation and all three dimensions of MCC were slightly higher than expected. However, the differences were negligible for the content and structure sub-scales and still small for the building relationship sub-scale (i.e. $\Delta r = .164$ between the expected and the observed correlation). These somewhat higher correlations might be explained by the fact that people with high social-organisation tend to maintain emotional balance, act calmly and with control, and adapt flexibly to changing conditions. All these factors are important when conducting a physician-patient conversation. In summary, we believe that the discussed deviations from our expectations are of marginal size and, thus, pose no substantial concern about discriminant validity against social competence.

Finally, the pattern of findings largely corroborated our expectations for the correlations with the Big Five personality traits. As a slight deviation, some of the correlations of conscientiousness and agreeableness with the VA-MeCo were marginally higher than $r = .1$; however, they can still be judged as essentially low correlations in terms of Cohen's [69] standards. As mentioned previously, the Big Five often show medium correlations with SJT performance [26]. The correlations found in this study were smaller throughout, indicating that personality plays a lesser role for performance on the VA-MeCo than is typically expected for SJTs.

In summary, despite the slight deviations discussed, the analyses provided a pattern of findings that was largely in line with our expectations (cf. Table 1) and delivers further evidence of the VA-MeCo's validity in measuring MCC [56,60]. The results support the conjecture that MCC is a hierarchical, multidimensional construct [49,50] that is related to but discernible from other factors that are frequently considered important in medical education (such as empathy and patient orientation) and, thus, provides added value.

## Limitations and future research

We acknowledge several main limitations of the present study. First, regarding factorial validity, some tasks (particularly Item 5) showed relatively low factor loadings. We retained these items because they represent important steps in medical communication and excluding them would have compromised the instrument's content validity. Nonetheless, as the overall factor solution is well-defined, theoretically coherent, and yields reliable scales, we consider it defensible, while noting that these tasks may warrant refinement in future research.

Second, we used a classic construct validity assessment approach [56,57,60] which has at least two limitations: (a) the selection of relevant validation variables and (b) the interpretation of correlation patterns, which is ultimately judgmental and somewhat subjective [83]. Regarding the first problem, we aimed to select the most relevant validation measures by drawing on substantive theory and research on communication competence, particularly in the medical field, as well as general research on the SJT method. Although we were able to include a fairly wide range of cognitive and non-cognitive variables, this selection is necessarily incomplete. However, the inclusion of further validation measures—particularly further achievement tests—would have meant an even greater workload for the participants, potentially compromising the quality of the data. For the same reason and given the sample size requirements for the analyses of the present study, the collection of additional behavioural measures of MCC (e.g., in an OSCE) was impractical and beyond the scope of the study.

Regarding the interpretation of correlations, we attempted to reduce subjectivity by stating a priori hypotheses about the expected directions and, where possible, the sizes of the correlations (Table 1), and by following Cohen's [69] standards for interpreting effect sizes. However, these recommendations should not be taken as universally valid and unambiguous cut-off values. For example, in Fig 1, we categorised the correlation coefficients for social competence and the Big Five personality variables as deviating from the hypotheses when their absolute values exceeded $r=.1$ (i.e., a small effect). However, as noted above, many of the correlations found should still be interpreted as small (e.g., $r=.16$ between agreeableness and relationship). Moreover, the present data do not allow us to explore the causes of deviations in correlation sizes from the expected values. Future research should investigate whether such patterns replicate across different validation measures.

Furthermore, although our study provides an in-depth evaluation of the factorial and construct validity of the VA-MeCo, further validation is needed, particularly studies assessing its predictive validity for communicative quality in actual patient encounters. Further research should examine whether students who score well on the VA-MeCo perform better in communication in clinical settings. Similarly, studies should investigate the relationship between the VA-MeCo and performance in simulated interactions (e.g. OSCEs), which was not feasible in the present study. Such research should account for the fact that SJTs reflect knowledge-based judgment rather than actual behaviour. Therefore, we expect the VA-MeCo to be most predictive of situation-specific judgment and decision making, such as recognising meaningful patterns, identifying critical moments in conversations, evaluating courses of action, and analysing and evaluating communicative situations [34]. Furthermore, as the VA-MeCo primarily addresses cognitive aspects, further research is needed to examine how these interact with procedural and attitudinal dimensions of MCC (such as verbalising and regulating emotions to convey empathy) and develop combined approaches that capture these facets more directly.

Additionally, the stability of MCC over time and its sensitivity to instructional interventions should be investigated. Longitudinal research could assess medical students with the VA-MeCo at multiple points during their studies to track changes in MCC and evaluate the instrument's sensitivity to communication training. Moreover, examining external and predictive validity by testing associations with performance in both simulated and real patient interactions would further strengthen the instrument.

Finally, a common criticism of SJTs is their potential susceptibility to socially desirable responses [84]. Such response tendencies can bias correlations with other variables [57] potentially affecting our analyses of convergent and discriminant validity. However, we believe that the VA-MeCo's test format largely mitigates social desirability bias. In each task, rather than selecting a single best response, participants must rate the effectiveness of every presented answer option on

three dimensions for achieving a given communication goal. This format prevents participants from simply choosing the most socially desirable option. Furthermore, even if participants' effectiveness ratings are influenced by social desirability considerations, they can only earn test points if their responses match the experts' effectiveness ratings. Overall, these design features enhance the robustness of the VA-MeCo against social desirability bias and strengthen the validity of the results. Nonetheless, future research should also examine other potential threats to validity, such as group-related biases (e.g., differential item functioning by gender or ethnic background). Investigating such aspects was beyond the scope of the present study due to the sample size requirements.

## Practical implications

The present study adds to the existing evidence for the validity of the VA-MeCo in measuring medical students' MCC in patient encounters. Due to its digital implementation and ease of use, the test can be administered effectively to large groups in medical education. The VA-MeCo is versatile and can be used for both assessment and teaching MCC. In the classroom, faculty can use the test to monitor student progress, provide feedback, and raise awareness of important medical communication issues. In many teaching scenarios, the VA-MeCo can be effectively combined with more resource-intensive methods, such as OSCEs and standardised patients, at different stages of the curriculum. Test results can provide valuable feedback to faculty, supporting curricular quality development. Finally, given its close alignment with standards for medical communication training [47,48], the VA-MeCo's materials may serve as inspiration for creating new learning opportunities or refining existing ones.

## Supporting information

**S1 Fig. Example of a test task.**
(TIF)

**S2 Fig. Path diagram with standardized results for M4.**
(TIF)

**S1 Table. Details of the instruments used for measuring validation variables.**
(DOCX)

**S2 Table. Standardised factor loadings and factor correlations of the final three-dimensional CFA-model (M4).**
(DOCX)

**S3 Table. Model fit of alternative one- and two-dimensional CFA models compared to the three-dimensional target model of MCC (M4).**
(DOCX)

## Acknowledgments

We thank all the medical students that participated in this study as well as our student assistants for their help in gathering the data.

## Author contributions

**Conceptualization:** Sabine Reiser, Eva Thomm, Johannes Bauer.

**Data curation:** Sabine Reiser, Laura Schacht.

**Formal analysis:** Sabine Reiser, Laura Schacht, Johannes Bauer.

**Funding acquisition:** Pascal O. Berberat, Martin Gartmeier, Johannes Bauer.

**Investigation:** Sabine Reiser, Laura Schacht.

**Methodology:** Sabine Reiser, Laura Schacht, Eva Thomm, Johannes Bauer.

**Project administration:** Sabine Reiser, Kristina Schick.

**Software:** Eva Dörfler.

**Supervision:** Pascal O. Berberat, Martin Gartmeier, Johannes Bauer.

**Visualization:** Sabine Reiser.

**Writing – original draft:** Sabine Reiser.

**Writing – review & editing:** Laura Schacht, Eva Thomm, Laura Janssen, Sylvia Irene Donata Pittroff, Kristina Schick, Pascal O. Berberat, Martin Gartmeier, Johannes Bauer.

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
