## [Decision Letter · Decision Letter 0]

22 May 2025

Dear Dr. Bauer,

Thank you for submitting your manuscript to PLOS ONE. After careful consideration, we feel that it has merit but does not fully meet PLOS ONE’s publication criteria as it currently stands. Therefore, we invite you to submit a revised version of the manuscript that addresses the points raised during the review process.

We look forward to receiving your revised manuscript.

Kind regards,

Ayesha Fahim

Academic Editor

PLOS ONE

Reviewers' comments:

Reviewer's Responses to Questions

**Comments to the Author**

1. Is the manuscript technically sound, and do the data support the conclusions?

Reviewer #1: Yes

Reviewer #2: Partly

Reviewer #3: Yes

Reviewer #4: Yes

Reviewer #5: Yes

2. Has the statistical analysis been performed appropriately and rigorously?

Reviewer #1: Yes

Reviewer #2: Yes

Reviewer #3: Yes

Reviewer #4: Yes

Reviewer #5: Yes

3. Have the authors made all data underlying the findings in their manuscript fully available?

Reviewer #1: Yes

Reviewer #2: Yes

Reviewer #3: Yes

Reviewer #4: Yes

Reviewer #5: Yes

4. Is the manuscript presented in an intelligible fashion and written in standard English?

Reviewer #1: Yes

Reviewer #2: Yes

Reviewer #3: Yes

Reviewer #4: Yes

Reviewer #5: Yes

Reviewer #1: As the authors identify, professionalism and in particular communication skills are some of the most important and yet notoriously difficult skills to teach to medical students.

Perhaps the first step to teaching or learning anything is to define what it is that we are attempting to teach or learn. This study goes a long way towards clarifying what it is we mean when we say "Medical Communication Competence".

Any step which bringings us closer to the development of a means to accurately measure these skills without bias provides feedback that can be used to improve students' communication skills and teachers' ability to teach these skills. By validating the constructs that underlie MCC this study supports the future development of more scaffolded approaches to teaching and learning MCC.

I would note that the article was information dense and that it took me several reads to unpack the study itself and the conclusions that the authors' had drawn. I would suggest the incorporation of simple visual aids to assist readers to understand the core concepts more easily... Perhaps Radar Diagrams or the like showing the factors and their correlation coefficients might be useful.

Reviewer #2: In the background include a paragraph that addresses what is currently happening with the measurement of communication competence in health sciences students, that is, whether current teaching methods and/or didactis are effective or why they are a challenge, where they are failing, or why current methods are not working.

Also, justify why a scale or test allows for the measurement of communication competence, which is an aspect that requires procedural and attitudinal skills, not just cognitive skills or decision-making or judgment.

Improve the wording of lines 84 and 92, as it currently appears contradictory. First, they state that there is a growing interest in the use of situational judgment tests (SJTs), and then in line 92, they mention "still only a few SJTs."

Methodology

It is suggested included the number of students who did not participate and indicated the reasons.

Please clarify if the 45 minutes were spent administering only the Va-MeCo or all the tests. Please also indicate how many questions were included in the total for all the tests, and what the total number of questions was. Although the authors mention that the information is reviewed according to references, please also mention if everything was self-administered or if there was a tutor who guided them through all the tests, or only the Va-MeCo.

All this information is necessary to allow other appropriately trained researchers to fully replicate your study.

Results

This section provides part of the discussion, as it not only describes the results but also briefly analyzes them. This is permitted by the journal; however, it is suggested that you create a combined Results/Discussion section (commonly labeled "Results and Discussion") or leave it as results only without providing additional information or expressing your opinion.

Discussion

Include references between lines 385 and 407 that support or contrast your results. For example, it is not clear why it allows for better feedback, or how it relates to feedback. Evidence that supports this or mentions its importance in the process is missing.

Limitations/Conclusion

Include in one of these sections, that communication competency is a procedural and attitudinal competency where students must verbalize and manage their emotions to empathize. Therefore, the current proposal is a cognitive approach, since the interaction of these facets requires further research and other types of instruments/methodologies for its measurement.

Reviewer #3: This manuscript “Assessment of medical communication competence“ presents a carefully conducted empirical study. The strengths of the manuscript lie in its conceptual clarity, the systematic approach to empirical analysis. It is well formulated in terms of language.

The authors describe an empirical analysis of the factorial and construct validity of an online situational judgement test (VA-MeCo), measuring medical students´ communication competence. They provide a detailed and well-structured background on the test and its conceptual foundations. The research question is clearly defined, and the hypotheses are quantified based on empirical findings. The methodology of confirmatory factor analysis seems appropriately selected and transparently described. Construct validity is operationalised and tested using convergent and discriminant variables. The discussion section effectively integrates the results with relevant literature and outlines implications in a differentiated manner.

However, the conclusion section stands out from the rest of the manuscript in terms of focus and content. Rather than providing a concise summary of the study’s aims, methods, and main findings, it introduces new aspects that have not been sufficiently addressed in earlier sections. While these points appear clearly relevant and valuable, they would benefit from being integrated into the introduction and discussion to ensure a coherent overall argument and to support the internal consistency of the manuscript.

The study addresses a topic of clear relevance within the field. It enriches the ongoing academic discourse and opens avenues for further research. Overall, the manuscript represents a valuable scholarly contribution and demonstrates a high standard of academic rigor.

Reviewer #4: The authors report a rigorous extension of the characterization of an instrument to measure an individual's skills in communication with a patient. The manuscript significantly expands the characterization beyond previous reports. The article would have more impact if it reported changes in learners' communication skills in response to an intervention. In terms of the quality of the validation, two tasks (tasks 1 and 5) fail to show standardized factor loadings above 0.5 (a commonly used cut-off) for any of the three subscales.

Reviewer #5: Reviewer comment and suggestions

This study on the Video-Based Assessment of Medical Communication Competence (VA-MeCo) presents valuable insights into measuring medical communication competence (MCC) among medical students. However few issues need improvement.

Strengths:

• Relevance of the Topic: The study addresses a critical area in medical education, emphasizing the importance of effective communication in patient encounters, which is well-documented in literature.

• Methodological Rigor: The use of confirmatory factor analysis (CFA) to assess factorial validity adds a strong methodological foundation, providing statistical support for the three-dimensional structure of MCC.

• Reliability: The high McDonald’s Ω values indicate that the VA-MeCo test is reliable, reinforcing its potential utility in educational settings.

• Construct Validity Exploration: Evaluating both convergent and discriminant validity enhances the understanding of how MCC relates to other cognitive and personality traits, suggesting that the VA-MeCo measures a specific and relevant construct.

• Implications for Medical Education: The findings support the integration of structured assessments like VA-MeCo into medical curricula, highlighting their role in improving the competence of future physicians.

Areas for Improvement:

• Sample Diversity: Although the sample size of 395 is substantial, the study should discuss the demographic diversity of participants. If the sample lacks variety in terms of age, gender, ethnicity, or educational background, this may limit the generalizability of the findings.

• Exploration of Deviation in Correlations: While the study mentions that correlation sizes partly deviated from expectations, it does not thoroughly explore the implications of this finding. A deeper analysis could provide insights into why certain relationships were weaker than anticipated and how this might inform future research.

• Longitudinal Validation: The study recommends further validation, but it would strengthen its claims to propose specific longitudinal or cross-validation studies. Demonstrating stability of MCC over time and its predictive validity concerning actual patient interactions would be beneficial.

• Practical Application: Though the study indicates that VA-MeCo is useful for assessing MCC, it could provide more detailed recommendations on how the findings can be implemented in medical training programs. For example, how should educators modify curricula based on VA-MeCo results?

• Potential Confounders: While the study suggests that VA-MeCo isn't confounded by traits like intelligence, it could be worthwhile to explore other potential confounding variables that may influence MCC and how these were controlled in the study.

NB:

Generally, this study is a strong contribution to the field of medical education, offering a rigorously validated instrument for assessing communication skills. Addressing the areas for improvement could enhance its robustness and applicability further, making it a more powerful tool for educators seeking to improve medical communication competence among students.

**Do you want your identity to be public for this peer review?** For information about this choice, including consent withdrawal, please see our Privacy Policy

Reviewer #1: **Yes: ** Francesco Bolstad

Reviewer #2: **Yes: ** Claudia Bugueno Araya

Reviewer #3: No

Reviewer #4: No

Reviewer #5: No

---

## [Author Response · Author response to Decision Letter 1]

30 Aug 2025

A detailed point-by-point response to the editor’s and reviewers’ comments can be found in the document Response to Reviewers.

---

## [Editor Report · Decision Letter 1]

7 Sep 2025

Online assessment of medical students' communication competence in patient encounters: Validation of the VA-MeCo situational judgement test.

PONE-D-25-19255R1

Dear Dr. Bauer,

We’re pleased to inform you that your manuscript has been judged scientifically suitable for publication and will be formally accepted for publication once it meets all outstanding technical requirements.

Kind regards,

Ayesha Fahim

Academic Editor

PLOS ONE